# Ultra-Violet-Assisted Scalable Method to Fabricate Oxygen-Vacancy-Rich Titanium-Dioxide Semiconductor Film for Water Decontamination under Natural Sunlight Irradiation

**DOI:** 10.3390/nano13040703

**Published:** 2023-02-12

**Authors:** Mohammed Alyami

**Affiliations:** Physics Department, College of Science and Humanities in Al-Kharj, Prince Sattam Bin Abdulaziz University, Al-Kharj 11942, Saudi Arabia; m.alyami@psau.edu.sa

**Keywords:** sunlight, oxygen vacancy, TiO_2_, photocatalytic activity

## Abstract

This work reports the fabrication of titanium dioxide (TiO_2_) nanoparticle (NPs) films using a scalable drop-casting method followed by ultra-violet (UV) irradiation for creating defective oxygen vacancies on the surface of a fabricated TiO_2_ semiconductor film using an UV lamp with a wavelength oof 255 nm for 3 h. The success of the use of the proposed scalable strategy to fabricate oxygen-vacancy-rich TiO_2_ films was assessed through UV–Vis spectroscopy, X-ray photoelectron spectroscopy (XPS), X-ray diffraction (XRD), and scanning electron microscopy (SEM). The Ti 2p XPS spectra acquired from the UV-treated sample showed the presence of additional Ti^3+^ ions compared with the untreated sample, which contained only Ti^4+^ ions. The band gap of the untreated TiO_2_ film was reduced from 3.2 to 2.95 eV after UV exposure due to the created oxygen vacancies, as evident from the presence of Ti^3+^ ions. Radiation exposure has no significant influence on sample morphology and peak pattern, as revealed by the SEM and XRD analyses, respectively. Furthermore, the photocatalytic activity of the fabricated TiO_2_ films for methylene-blue-dye removal was found to be 99% for the UV-treated TiO_2_ films and compared with untreated TiO_2_ film, which demonstrated only 77% at the same operating conditions under natural-sunlight irradiation. The proposed UV-radiation method of oxygen vacancy has the potential to promote the wider application of photo-catalytic TiO_2_ semiconductor films under visible-light irradiation for solving many environmental and energy-crisis challenges for many industrial and technological applications.

## 1. Introduction

The environmental hazard posed by gaseous pollution has recently raised serious concerns [1,2,3,4,5,6]. Specifically, several harmful health-related and environmental issues emanate from NO_x_-based pollutants, while various measures to control these pollutants revolve around the fabrication of photo-catalyst with desirable features and the ability to oxidize harmful pollutants for environmental remediation [7,8,9,10]. Photocatalyst-coated surface-based reactors have proven to be more practical for long-term operation than photocatalytic powder-based reactors [11,12,13,14]. As a promising photo-electrode and photocatalyst, titanium dioxide (TiO_2_) has enjoyed wider applicability in photocatalytic hydrogen generation, solar cells, and the remediation of organic contaminants among other photo-catalytic applications [15,16,17]. Furthermore, TiO_2_ is recognized as a low-cost, highly effective and photo-catalyst of interest as a result of its promising thermal and chemical stabilities, desirable electronic features, and environmental benignity, among others [18,19,20,21]. Pristine TiO_2_ semiconductors are characterized by a wide band gap that can only utilize the UV part of the light spectrum with wavelengths shorter than 385 nm, which is just 5% of the sunlight energy capacity. The extension of spectrum usability to visible regions requires further and more extensive research [22,23,24,25]. Additionally, the rapid recombination of photo-generated holes and electrons further restricts the practical applicability of these semiconductors [26,27]. Previous theoretical and experimental efforts to extend the separation period of photo-generated carriers and to reach a narrower band gap included the formation of defects, metal and non-metal doping, hydrogenation, noble-metal deposition, defect engineering, sensitization, and hetero-junction formation, among others [28,29]. Oxygen vacancy potentially modulates the semiconductor band gap and influences the band properties and structure [30,31,32]. Such defects as vacancies also improve charge-carrier-separation efficiency through electronic-conductivity improvement [33]. Similarly, oxygen vacancies conveniently modify the electronic structure in the reaction site’s vicinity and eventually facilitate intermediate adsorption, while the photo-catalytic activity is improved [34,35,36,37]. Oxygen vacancies are widely employed in defect formation in photo-catalysts for enhancing the photocatalytic activity of TiO_2_ semiconductors. The experimental characterization and theoretical computations in the literature revealed the creation of a mid-gap state below the conduction band due to the oxygen vacancies incorporated in the semiconducting materials, as well as the generation of Ti^3+^ centers. These centers prevent rapid electron-hole recombination while the energy gap is lowered by the mid-gap states created by the oxygen vacancies. Therefore, the photocatalytic activity of TiO_2_ semiconductors can be effectively enhanced through oxygen-vacancy creation. Immense efforts were deployed in order to achieve the synthesis of oxygen-vacancy-mediated TiO_2_ semiconductors for addressing several photo-catalytic challenges [4,38,39]. Current methods and techniques for incorporating and controlling oxygen vacancies in TiO_2_ semiconductors include thermal treatment with hydrogen, oxygen-depletion-based thermal treatment, self-doping, particle bombardment using high energy, and UV irradiation [4,33,38,39]. This work proposes the UV-irradiation method of oxygen-vacancy creation, which is characterized by simplicity, effectiveness, and versatility compared to other methods, such as the hydrothermal technique. The created oxygen vacancies were assessed using various spectroscopic techniques, including XPS and UV–Vis, among others, to reveal the relation between photocatalytic performance and the presence of oxygen vacancy.

## 2. Experimental Section

### 2.1. Sample Preparation

The TiO_2_ nanoparticles (<25 nm, 99.5%, Sigma Aldrich, USA) were mixed with ethanol to obtain a concentration of 5 mg/L using a magnetic stirrer. The stirring lasted 30 min at 500 rpm, while the drop-casting technique was employed for applying TiO_2_-nanoparticles dispersion on a glass substrate. The glass substrate was subsequently placed on a hot plate at a temperature of 80 degrees Celsius for ethanol removal. This eventually produced a solid film of TiO_2_ nanoparticles on the glass substrate. Two different substrates were synthesized using aforementioned experimental conditions, while one substrate with solid TiO_2_ nanoparticles was exposed to UV radiation of 255 nm at a distance of 15 cm from a 6-watt UV lamp (UVP UVGL, Analytik Jena, Germany) while the second substrate was not treated with UV exposure. The schematic diagram illustrates the processes and an experimental procedure is presented in Figure 1.

### 2.2. Characterization

Using an X-ray diffractometer (A Shimadzu XRD-6000, Kyoto, Japan) operating in the range 10° ≤ 2θ ≤ 80°, the crystal structure of the as-prepared TiO_2_ film was examined. The morphology of the as-prepared TiO_2_ film was examined by using scanning electron microscopy (FESEM, Quanta FEG250, FEI, Hillsboro, OR, USA). Furthermore X-ray photoelectron spectroscopy (XPS; Thermo scientific K-alpha XPS spectrometer, Thermo Fisher Scientific, Waltham, USA) was used to distinguish and characterize the chemical makeup of the as-prepared TiO_2_ films. A monochromic Al ka source with a characteristic energy of 1486.6 eV was utilized. A spectrophotometer was used to log the UV–Vis absorption spectra of the as-prepared TiO_2_ films. 

### 2.3. Photocatalytic Activity 

By making use of a constructed immobilized photocatalytic reactor [33] fitted with a magnetic stirrer working at 500 r.p.m., the photocatalytic performance of the as-prepared TiO_2_ films ~ 3 cm^−2^ was investigated for an aqueous solution of methylene-blue dye (~5 ppm) under solar radiation. The setup was left wholly in the dark for about 60 min before each experiment to allow the adsorption–desorption equilibrium to be reached. The as-prepared sample was then placed in sunlight in Al Kharj City, Saudi Arabia, at noon. From the methylene-blue-dye absorption spectra over 300–750 nm, the ratio of methylene-blue-dye concentration at t min (Ct) to methylene-blue-dye concentration at 0 min (C0) was calculated at regular time increments of ~15 min.

## 3. Results and Discussion 

The bandgap and optical absorption of the readymade TiO_2_ film in its primal and UV-irradiated state were investigated via UV-Vis diffuse-reflectance spectroscopy. From this analysis, it was observed that the UV-treated TiO_2_ films had a higher absorption in the visible region than the untreated TiO_2_ films (Figure 2a). In terms of the bandgap energy, the UV-treated TiO_2_ films demonstrated a narrower bandgap of approximately 2.95 eV, compared to the 3.2 eV observed in the untreated TiO_2_ films (Figure 2b). Valence-band energy can be obtained from XPS valance-band spectra through extrapolation to the binding-energy axis. The energy of a valence band is the energy of the band of electron orbitals with which electrons jump out (and move into the conduction band) when excited. The determination of the valence-band energy from the acquired XPS valence-band spectra is presented in Figure 2c,d. The valence-band energy of the TiO_2_ that was not exposed to UV radiation is presented in Figure 2c, while that of the TiO_2_ sample subjected to UV radiation for possible oxygen-vacancy creation is presented in Figure 2d. The valence-band energy for both the UV-treated and the untreated TiO_2_ film had a similar value because the top of the valence band was governed by the O 2p states, while the bottom of the conduction band was controlled by the Ti 3d state. Therefore, a defect (in Ti^3+^) that formed just below the conduction band was responsible for the variation in the conduction-band energy for the UV-treated and untreated TiO_2_ films. Hence, the UV-treated sample was characterized by a reduced/narrow energy gap after the UV exposure.

### Photocatalytic Performance

To experimentally verify the results obtained from the optical characterization analysis, the rate of the photodegradation (Ct/C0) of the methylene-blue dye by the TiO_2_ films was measured under the sunlight irradiation, as shown in Figure 3a. An outstanding photodegradation rate of approximately 99% was reached in 60 min using the UV-treated TiO_2_ film, which considerably surpassed the 77% photodegradation rate yielded by the untreated TiO_2_ film under the same operating conditions. 

Using the experimental results, the photocatalytic mechanism of the readymade UV-treated TiO_2_ film with oxygen vacancies was formulated. Figure 3b is a schematic diagram of the photocatalytic degradation of the methylene-blue dye on the UV-treatedTiO_2_ film. Electron-hole pairs were produced by the photoexcitation of electrons from the valence band (VB) into the conduction-band (CB)/oxygen-vacancy energy level upon the exposure of the UV-treated TiO_2_ film to UV-visible light irradiation. The photo-generated electrons were easily trapped by the oxygen vacancies, resulting in a low recombination rate with holes. Consequently, the electrons lived longer and reduced the amount of oxygen assimilated from the surface of the UV-treated TiO_2_ film and created superoxide radicals (**^•^**O_2_^−^), which are potent oxidizers of methylene-blue-dye molecules [40,41]. Meanwhile, photo-generated holes also spread out to the surface of the UV-treated TiO_2_ film and further oxidized any surface-assimilated methylene-blue dye. Consequently, the UV-treated TiO_2_ film with oxygen vacancies demonstrated superior visible-light photocatalytic performance. Generally, the oxygen vacancy defects on the surface of the photocatalyst enhance the separation efficiency of electron-hole pairs and ensure impeccable photocatalytic efficiency [17].

The XPS survey spectra acquired from the two prepared TiO_2_ samples are presented in Figure 4. The spectra were analyzed using Avantage software (version 5.932, Thermo Scientific, Waltham, MA, USA). The elemental identification and chemical states of the prepared samples, along with the binding energies, are shown in Figure 4a,b. Three different peaks were observed in the spectra of the untreated TiO_2_ semiconductor presented in Figure 4a, which correspond to carbon (C 1s) at a binding energy of 284.80 eV, titanium (Ti 2p3/2) at a binding energy of 458.10 eV, and oxygen (O 1s) band at a binding energy of ~531 eV. However, when the samples were treated with an ultra-violet (UV) beam for oxygen-vacancy creation, similar constituent element peaks and binding-energy positions were observed and are presented in Figure 4b. The carbon signal (C 1s) was observed at a binding energy of 284.53 eV, with a positive binding-energy shift of 0.66 eV due to exposure to the UV beam. The shift in the carbon-containing compound signal was due to the surface-cleaning potential of the UV beam. Furthermore, the titanium Ti 2p state was observed at the 458.94 eV [40] binding-energy peak with a positive binding-energy shift of 0.82 eV due to the possible formation of some new states of titanium after UV exposure. The new states that appeared after the UV treatment were vividly clear in the high-resolution spectra of the Ti 2p signal. The signal corresponding to oxygen O 1s appeared at 530.13 eV for the UV-treated sample, as shown in Figure 4b. The appearance of the new chemical state of oxygen further resulted in a positive binding -energy shift of 0.63 eV as compared to the untreated sample. Table 1 presents the survey -spectra parameters for the sample exposed to UV, as well as the untreated samples. The binding energy and the atomic percentage of the samples before and after UV exposure are also presented in Table 1. The peak areas of each element, except the C 1s band, is increased after exposure to UV light. The atomic percentage of the Ti 2p and O 1s increased after the UV exposure, while that of the carbon C 1s decreased after UV exposure. This indicates the contaminant -removing potentials of UV light.

The high-resolution spectra of the Ti 2p doublet for the untreated and UV-treated TiO_2_ film samples are shown in Figure 4c,d. Two peaks were identified in the spectra of the untreated sample, as shown in Figure 4c, and they were attributed to Ti^4+^ 2p_3/2_ and Ti^4+^ 2p_1/2_ components. The components were located at binding energies of 458.12 and 463.74 eV, respectively [41]. The energy difference obtained for the doublet components for the untreated TiO_2_ sample was 5.7 eV, which shows the presence of an anatase phase in the TiO_2_ sample [42]. The spectra for the UV-treated samples are presented in Figure 4d, with four different peaks at different binding energies. The dominant Ti^4+^ 2p_3/2_ and Ti^4+^ 2p_1/2_ lines of titanium present in the untreated sample were maintained with binding energies of 458.48 and 464.18 eV, respectively. This shows that the Ti^4+^ 2p_3/2_ and Ti^4+^ 2p_1/2_ states shifted positively due to the change in the surface-charging effect. It is worth mentioning that the anatase phase was maintained after UV exposure, as can be observed from the doublet value of 5.7 eV. Additional Ti^3+^ 2p_3/2_ and Ti^3+^ 2p_1/2_ states were exhibited at binding energies of 456.90 and 460.99 eV, respectively, and the energy difference was 4.09 eV. Initially, the presence of a Ti^3+^ signal indicates the presence of oxygen vacancies after Ti^4+^ ions undergo a reduction process. The XPS spectra of the oxygen O 1s signal for the untreated and UV-treated samples of TiO_2_ semiconductor films are presented in Figure 4e,f. For the untreated-sample spectra presented in Figure 4e, two peaks were exhibited at 529.21 and 531.07 eV [42], which correspond to lattice oxygen and surface-chemisorbed hydroxyl group (any other surface-oxide species are also probable), respectively. For comparison, in the high–resolution-spectra UV-treated samples shown in Figure 4f, three different components are exhibited.

The peaks were located at 529.70, 531.77, and 531.22 eV, which correspond to lattice oxygen, surface-chemisorbed hydroxyl groups, and oxygen vacancies, respectively. The lattice-oxygen peak shifted positively by 0.49 eV, while the surface-chemisorbed hydroxyl group showed a positive shift of 0.70 eV, followed by the occurrence of an oxygen-vacancy peak. The observed binding-energy shift can be attributed to oxygen-vacancy formation, which facilitates electron transfer to Ti and O atoms. The electronic properties of the prepared samples were assessed using the binding-energy difference (BED) approach, which measures the binding-energy difference between O 1s and Ti 2p_3/2_ core levels [43]. Using this approach for the data presented in Figure 4c,e, BDE = BE (O 1s)-BE (Ti 2p_3/2_) = 529.21 eV −458.12 eV = 71.1 eV. The obtained BED of 71.1 eV shows that the employed peaks were from the Ti^4+^ states in TiO_2_. For the dominant components shown in Figure 4d,f, the BED of 71.2 eV also confirms the Ti^4+^ states in the TiO_2_. Using the minor components shown in Figure 4d,f, the BED of 74.32 eV confirms the Ti^3+^ states in TiO_2_.

The X-ray diffraction (XRD) patterns of the UV-treated and untreated TiO_2_ films are presented in Figure 5. The high level of similarity in the diffraction pattern suggests the minimal influence of UV radiation on the crystalline structure of the pure untreated TiO_2_ film.

The characterization of the UV-treated and untreated TiO_2_ films using scanning electron microscopy (SEM) is presented in Figure 6. The morphology of the TiO_2_ film after the UV exposure was insignificantly affected by the UV radiation.

## 4. Conclusions

In this work, a solid film of TiO_2_ nanoparticles was synthesized, and, further, UV radiation was employed for oxygen-vacancy creation to enhance the photocatalytic activity of the semiconductor under visible-light irradiation. Both the UV-treated and untreated samples were characterized using UV–Vis spectroscopy, XPS, XRD, and SEM. The survey XPS spectrum was acquired to show the presence of three constituent elements, carbon, titanium, and oxygen, with their respective chemical states indicated by the C 1s, Ti 2p, and O 1s lines. The high-resolution O 1s spectrum obtained from a TiO_2_ sample not exposed to UV contained two peaks, which were attributed to lattice oxygen and surface-chemisorbed hydroxyl groups. The binding energies of these two peaks shifted to higher positive values after the sample was exposed to UV radiation, confirming the presence of oxygen vacancies in the UV-radiated sample. The presence of an additional peak ascribed to oxygen vacancies in the spectra of the sample exposed to UV radiation further confirmed the versatility of the proposed UV-irradiation method for oxygen-vacancy creation. The presence of a Ti^3+^ oxidation state in the UV-treated sample due to the reduction of Ti^4+^ offers additional confirmation of the successful creation of oxygen vacancies. The band gap of the untreated TiO_2_ film was reduced from 3.2 to 2.95 eV after the UV exposure due to the oxygen vacancies created, which was made evident by the presence of Ti^3+^ ions. Radiation exposure has no significant influence on sample morphology or peak pattern, as revealed by the SEM and XRD analyses, respectively. During the methylene-blue dye removal, the UV-treated sample showed 99% capacity, while the untreated sample attained 77% capacity with the same operating conditions under natural-sunlight irradiation. The simplicity, scalability, and versatility of the proposed UV-radiation method of oxygen-vacancy creation can enhance and promote the photocatalytic activity of TiO_2_ semiconductor films for various desirable photocatalytic applications under solar-light irradiation.

## Figures and Tables

**Figure 1 nanomaterials-13-00703-f001:**
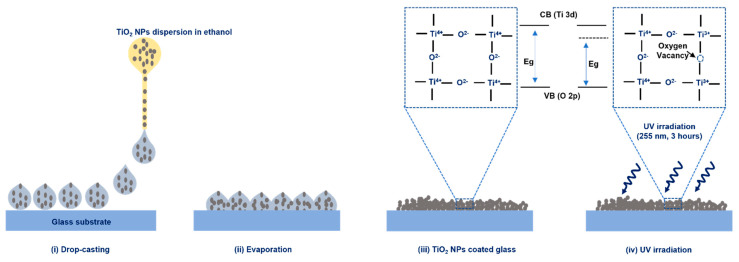
Fabrication strategy aimed at the fabrication of untreated and UV-treated TiO_2_ films.

**Figure 2 nanomaterials-13-00703-f002:**
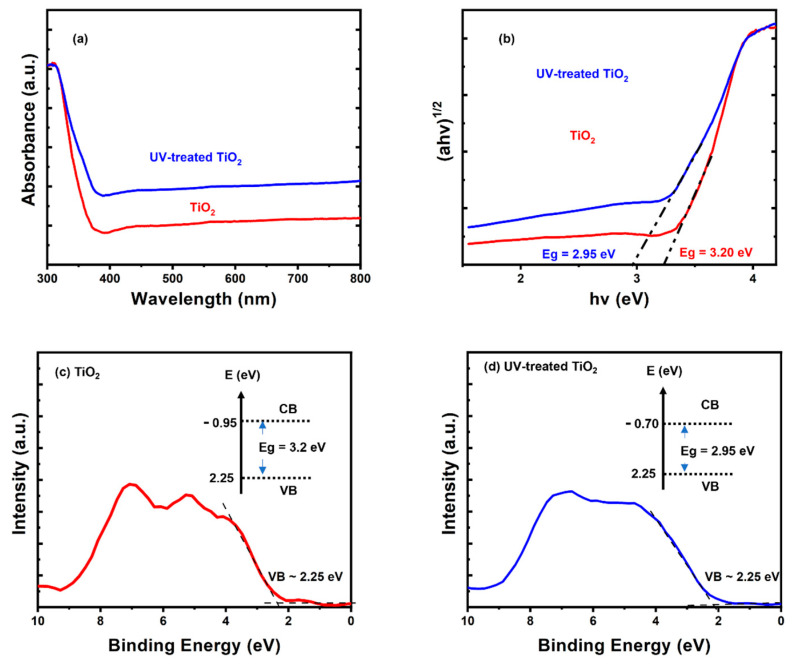
Optical characterization: (**a**) UVVis absorbance spectra, (**b**) estimation of the band-gap energies from Tauc’s plots, (**c**,**d**) XPS valence-band spectra of untreated and UV-treated TiO_2_ films, along with their electronic band structures, as indicated.

**Figure 3 nanomaterials-13-00703-f003:**
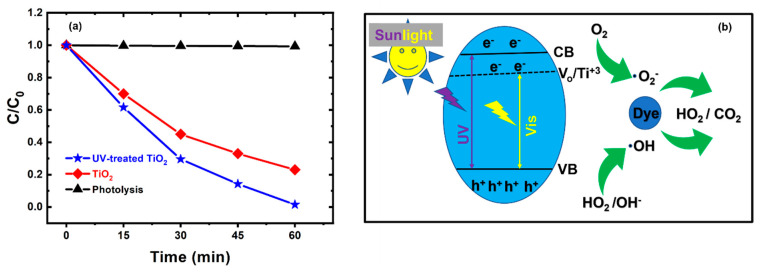
(**a**) Photocatalytic performance of the untreated and UV-treated TiO_2_ films. (**b**) Schematic illustration of the photocatalytic degradation of the methylene-blue dye on the UV-treated TiO_2_ film.

**Figure 4 nanomaterials-13-00703-f004:**
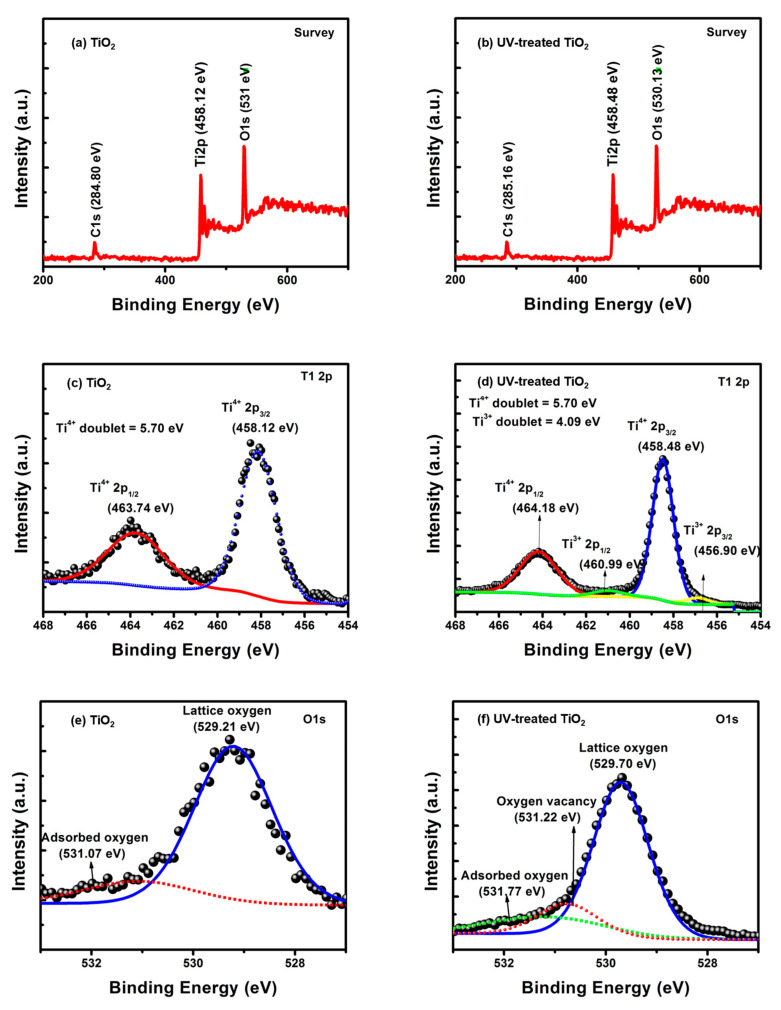
The XPS survey (**a**,**b**) and high-resolution XPS spectra of Ti 2p (**c**,**d**) and O 1s (**e**,**f**) of (**a**) untreated (**b**) UV-treated TiO_2_ films, respectively.

**Figure 5 nanomaterials-13-00703-f005:**
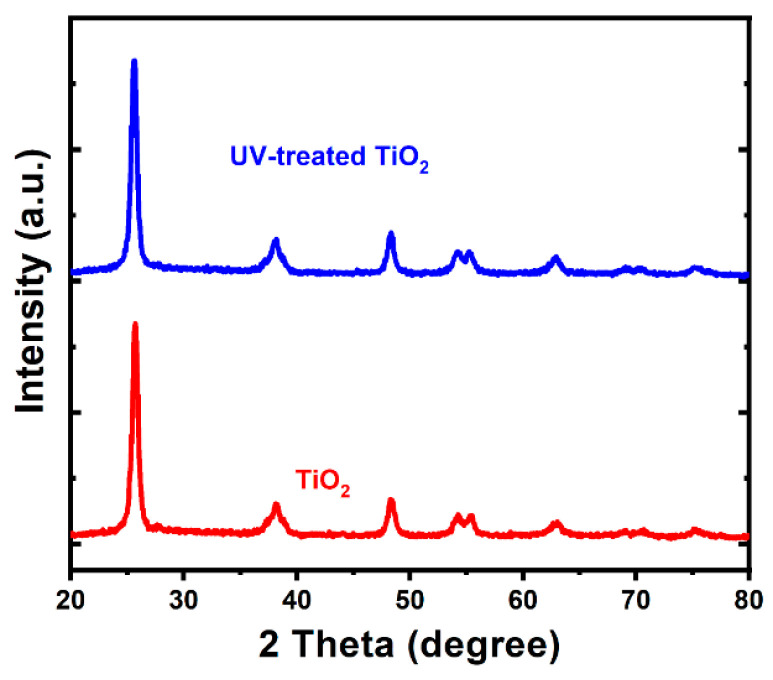
The XRD patterns of the untreated and UV-treated TiO_2_ films, as indicated.

**Figure 6 nanomaterials-13-00703-f006:**
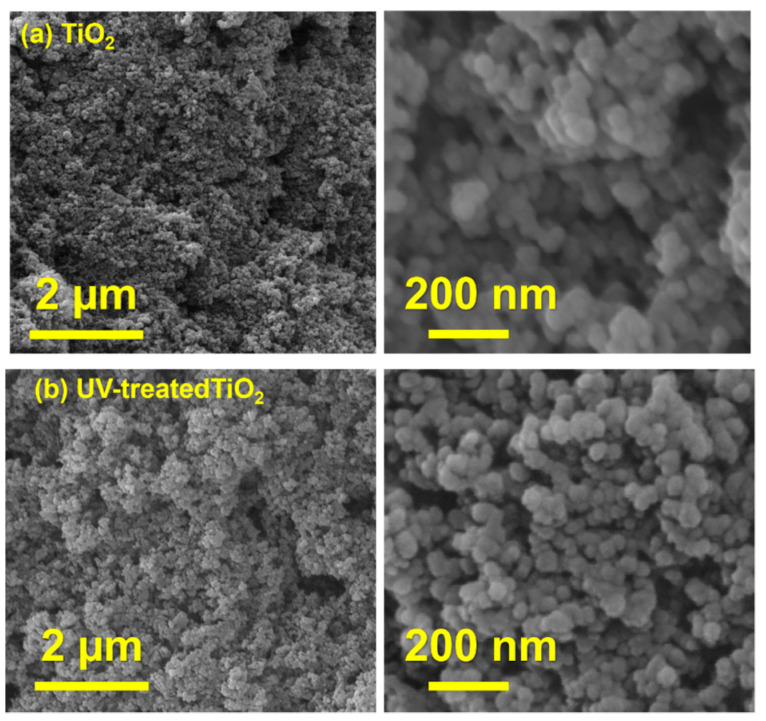
The SEM images of the untreated and UV-treated TiO_2_ films at different magnifications, as indicated.

**Table 1 nanomaterials-13-00703-t001:** Survey -spectra parameters for untreated and UV-treated samples.

Sample	Name	Peak BE (eV)	Atomic%
untreated	Ti 2p	458.20	23.38
-	O 1s	529.47	45.14
-	C 1s	284.53	31.47
UV -treated	Ti 2p	458.94	24.48
-	O 1s	530.13	50.06
-	C 1s	285.14	25.46

## Data Availability

The data presented in this study are available on request from the corresponding author.

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
