# Peer review of "Ultra-Violet-Assisted Scalable Method to Fabricate Oxygen-Vacancy-Rich Titanium-Dioxide Semiconductor Film for Water Decontamination under Natural Sunlight Irradiation"

_nanomaterials, 2023, doi:10.3390/nano13040703_

Round 1

Reviewer 1 Report

My comments are in attached file.

Reviewer 2 Report

This paper describes that oxygen vacancies were formed on TiO2 surface by UV irradiation and the obtained vacancy-rich TiO2 showed higher photocatalytic activity than the pristine TiO2 for photodegradation of methylene blue by sunlight. The generation of oxygen vacancies on TiO2 surface has been intensively studied including UV irradiation (e.g., Nano Lett. 2021, 8348). I could not find the what is advantage or new findings in this paper compared to the previous reports. Interpretation and discussion of XPS is also questionable. Therefore, I do not recommend this paper for publication in nanomaterials. Other comments are listed below.

1)     The author discussed the difference in binding energies between the vacancy-rich TiO2 and the pristine TiO2, but the peak was not corrected by the reference peak such as C 1s, Au 4f, etc. Because XPS peaks changes by each measurement even though the same sample is measured, the XPS must be referenced to C 1s of carbon tape, Au 4f by adding a small amount of Au foil on the sample, etc. Carbon is always contaminated. The discussion of the difference in electronic state of carbon does not make sense. In addition, even if the carbon contamination on the TiO2 surface is true, 25-30 % of C contamination is too high, and thus, the effect of oxygen vacancy would be reduced.

For XPS, the author mentioned that spin orbital splitting decreased to 4.09 eV for Ti(III) from 5.6-5.7 eV for Ti(IV) but the reason was not provided. If Ti(III) 2p3/2 was observed at 456.9 eV, the Ti(III) 2p1/2 peaks is supposed to be observed at around 462 eV. In addition, it seems there are small shoulder peaks in untreated TiO2; Ti 2p3/2 at 455.5 eV and O 1s shoulder peak at around 530.5 eV which were attributed to the Ti(III) species and oxygen vacancy, respectively. Thus, peak fitting seems to be inappropriate.

Reviewer 3 Report

The authors present a method to induce oxygen vacancy defects in TiO2 via UV irradiation, leading to enhanced photocatalytic methylene blue degradation. This is interesting and quite novel. However, the evidence that oxygen vacancies are actually formed need to be strengthened, in my opinion.

1. The authors should carefully review the literature on the different methods to produce oxygen vacancies in TiO2. In particular, has this method of UV irradiation for vacancy formation been reported before?

2. Fig 2a shows a large "absorbance" in the 400-800 nm range, indicating significant light scattering. This can make band-gap determination rather difficult. Are the authors able to produce a sample without scattering? Also, the plots in Fig 2a look quite similar, but for some reason the band-gap determination in Fig 2b is different. Are there any reports in the literature showing band-gap change with UV irradiation?

3. Fig 4d is supposed to show evidence for the existence of Ti3+, but the Ti3+ peaks are not visible to me. Are these peaks determined by software, or manually added? Can the software quantify the height of the Ti3+ peaks?

4. The authors claim that superoxide radicals are produced, without performing any scavenger experiments to prove this claim. Ideally, they should perform this experiment, or at the very least cite some literature to support this. 

Round 2

Reviewer 3 Report

The manuscript has improved after revision. I have just one suggestion: reference 33 is also a way to create oxygen vacancies in TiO2, but with UV light. This reference should be included in the introduction as a prior work introducing the creation of oxygen vacancies with UV. 

Author Response

Dear Professor,
Thank you for your effort to improve my manuscript!
The response to the reviewer's comment (Round 2) is attached here.
Dr. Alyamia
